# IsoBench: Benchmarking Multimodal Foundation Models on Isomorphic Representations

**Deqing Fu**$^{\psi *}$    **Ruohao Guo**$^{\delta *}$    **Ghazal Khalighinejad**$^{\lambda *}$    **Ollie Liu**$^{\psi *}$.
**Bhuwan Dhingra**$^{\lambda}$    **Dani Yogatama**$^{\psi}$    **Robin Jia**$^{\psi}$    **Willie Neiswanger**$^{\psi}$
$^{\psi}$USC    $^{\delta}$Georgia Tech    $^{\lambda}$Duke

## Abstract

Current foundation models exhibit impressive capabilities when prompted either with text only or with both image and text inputs. But do their capabilities change depending on the input modality? In this work, we propose **IsoBench**, a benchmark dataset containing problems from four major areas: math, science, algorithms, and games. Each example is presented with multiple **isomorphic representations** of inputs, such as visual, textual, and mathematical presentations. IsoBench provides fine-grained feedback to diagnose performance gaps caused by the form of the representation. Across various foundation models, we observe that on the same problem, models have a consistent preference towards textual representations. Most prominently, when evaluated on all IsoBench problems, Claude-3 Opus performs 28.7 points worse when provided with images instead of text; similarly, GPT-4 Turbo is 18.7 points worse and Gemini Pro is 14.9 points worse. Finally, we present two prompting techniques, *IsoCombination* and *IsoScratch-Pad*, which improve model performance by considering combinations of, and translations between, different input representations. Our dataset and evaluation code can be found at https://isobench.github.io/.

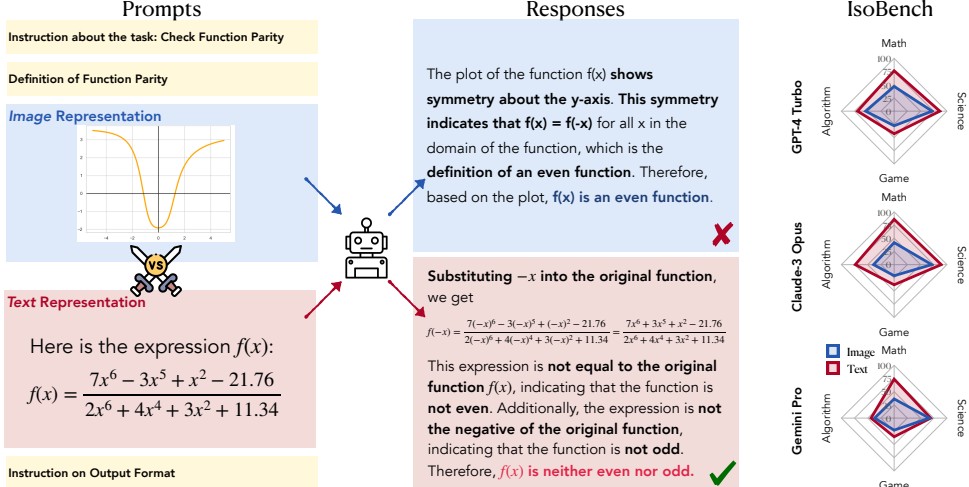

Figure 1: **Do multimodal foundation models treat every modality equally?** In this example, a model is provided with either an image representation or a text representation *isomorphic* to the image, where the instructions are kept identical. Surprisingly, multimodal models often give different responses for these isomorphic inputs (*e.g.*, in the figure above, only the response to the text representation is correct). In **IsoBench**, we scale such examples into four domains (*Math*, *Science*, *Algorithms*, *Games*) and find a consistent preference towards text across many popular multimodal foundation models.

## 1  Introduction

On the heels of the large language model (LLM) revolution, we are currently witnessing a second revolution of *multimodal foundation models*. Exemplified by models like GPT-

---

*Equal Contribution, alphabetical order. {deqingfu, yogatama, robinjia, neiswang}@usc.edu, {ghazal.khalighinejad, bhuwan.dhingra}@duke.edu, rguo48@gatech.edu, me@ollieliu.com

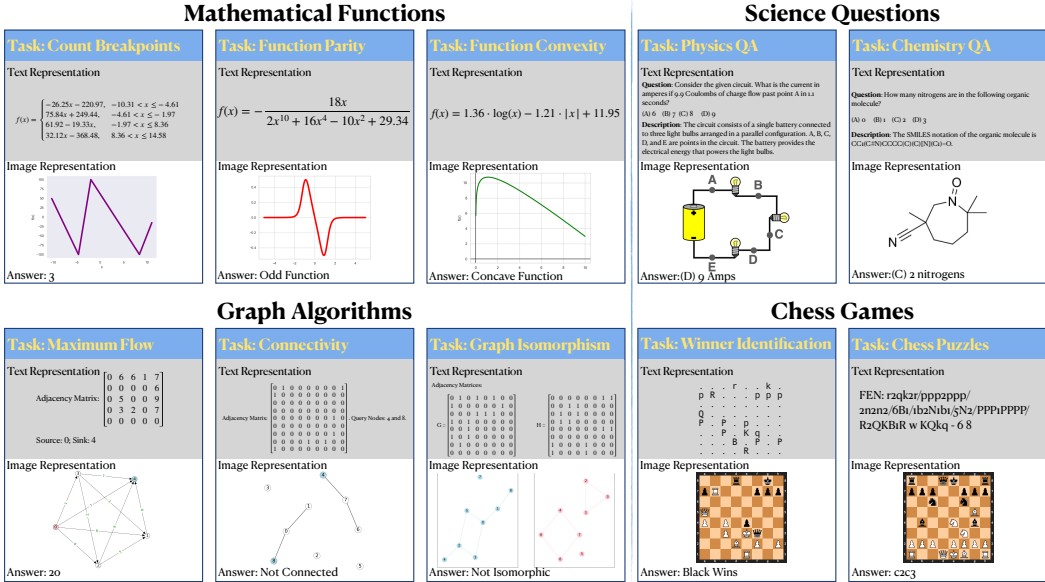

Figure 2: **IsoBench** contains four major domains: *Mathematical Functions*, *Science Questions*, *Graph Algorithms*, and *Chess Games*. For each domain, there are two or three subtasks. All examples within IsoBench are provided with one image representation and several textual representations that are isomorphic to each other.

4V (OpenAI, 2023b), Claude (Anthropic, 2024), and Gemini (Google, 2023), these models combine a large language model backbone with a vision encoder that enables them to accept either pure text inputs or a combination of text and images. The rapid rise of these multimodal models necessitates new benchmarks that accurately assess their capabilities.

In this work, we study whether these models exhibit the same capabilities when processing text and image inputs. Our key insight is that many problems consisting of both image and text prompts can be equivalently formulated with text-only prompts. By studying how well models do on both representations of the same problem, we can evaluate whether the multimodal fusion components of these models truly empower the model with the same capabilities when reasoning about images and text. While many benchmarks already exist for testing multimodal foundation models (Yue et al., 2023; Lu et al., 2024; Zhang et al., 2024a; Fan et al., 2024; Zhang et al., 2024b), none of these measure models' performance discrepancies between semantically equivalent inputs in different modalities.

We create *IsoBench*, a broad-coverage benchmark for evaluating multimodal foundation models on 10 different tasks with multiple isomorphic representations. IsoBench includes four major subjects: mathematical functions, algorithmic problems, science questions, and chess games. For each test example, IsoBench provides one image representation and one or more alternative textual representations. We evaluate a suite of multimodal foundation models, including GPT-4[1], Gemini, and Claude-3, on IsoBench, and find that all tested models perform substantially better when given textual representations compared with visual representations. This bias in favor of textual representations runs counter to human cognition: humans are known to exhibit a *picture superiority effect* (Defeyter et al., 2009), a preference towards visual representations, and humans generate visual images internally regardless of whether the required task is visual or verbal (Amit et al., 2017).

Finally, we design two simple mechanisms to further study the performance of multimodal foundation models on tasks with isomorphic representations: *IsoCombination* (IsoCB), which feeds multiple isomorphic representations to the model, and *IsoScratchPad* (IsoSP), which

---

[1]Throughout this paper, we use GPT-4 as a general term when referring to both the GPT-4 large language model, and the GPT-4V multimodal foundation model.

| Subject | Representation | GPT-4 Turbo | Gemini 1.0 Pro | Claude-3 Opus | Mixtral-8x7B | *Random Guess* |
|---------|----------------|-------------|----------------|---------------|--------------|----------------|
| Science | Image | **72.0** | 66.7 | 71.3 | – | *38.3* |
|         | Text | 86.7 | 69.3 | **89.3** | 68.0 | |
| Mathematics | Image | **46.9** | 36.4 | 41.7 | – | *44.4* |
|             | Text | 76.6 | 73.9 | **85.8** | 66.1 | |
| Algorithms | Image | **54.2** | 37.0 | 39.1 | – | *34.7* |
|            | Text | 69.5 | 43.8 | **73.7** | 49.5 | |
| Games | Image | **27.6** | 22.9 | 21.0 | – | *18.1* |
|       | Text | **42.8** | 34.9 | 33.4 | 35.5 | |
| Average | Image | **50.2** | 40.7 | 43.3 | – | *33.9* |
|         | Text | 68.9 | 55.6 | **72.0** | 57.2 | |
|         | Delta (Text – Image) | +18.7 | +14.9 | +28.7 | – | |

Table 1: **IsoBench results**. Scores are evaluated as accuracy. Within each subject, the best scores in processing image representations are highlighted in blue, and the best in text representations are in red. Overall, GPT-4 Turbo is the best for images and Claude-3 Opus is the best for text. Across four subjects within IsoBench, multimodal foundation models have a strong preference for text modalities. The gap in accuracy between the best text representation and its isomorphic image representation can be as large as 28.7%.

uses a scratchpad to translate between visual and text representations. We find that both methods improve aspects of multimodal models, with IsoCB improving model performance on graph algorithm problems by up to 9.4 points compared with the best single representation, and IsoSP improving performance on science problems by up to 14.4 points.

In full, we summarize our contributions as follows:

- We introduce IsoBench, a test dataset consisting of 1,887 samples[2] spanning diverse domains such as discrete and applied mathematics, physics, chemistry, and chess. For each sample, we evaluate multiple *isomorphic* input representations containing the same information—one with a visual representation and others with domain-specific textual representations—to facilitate multimodal performance assessments.

- We benchmark eight popular foundation models, and find that each of the tested multimodal models perform substantially better on text-only prompts than image-based prompts, in contrast with known human preferences for images over text.

- To bridge performance discrepancies between different input modalities, we introduce IsoCombination (IsoCB) and IsoScratchPad (IsoSP), two procedures that, respectively, fuse input modalities and transform visual input into textual representation at inference time. In certain settings, we find that IsoCB and IsoSP can improve the performance of multimodal foundation models by nearly 10 percentage points.

## 2  IsoBench

IsoBench is a collection of problems of the form $(\mathcal{P}, \{\mathcal{R}_1, \cdots, \mathcal{R}_m\})$, where $\mathcal{P}$ is a natural language instruction of task, and $\mathcal{R}_i$'s are different representations of the sample. Ideally, these representations are *isomorphic* to each other, *i.e.* for any pairs of input $\mathcal{R}_i, \mathcal{R}_j$, there exists a bijective function $\phi$ such that $\phi(R_i) = R_j$. As illustrated in Figure 1, the prompts to foundation models share the same instructions except for the problem descriptions (where one of the representations $\mathcal{R}_j$ is selected). We now detail the construction of IsoBench.

---

[2]The IsoBench dataset (www.huggingface.co/datasets/isobench/IsoBench) currently consists of 1,887 examples across four domains, though the construction of a larger dataset is currently underway.

| | | GPT-4 Turbo | GPT-3.5 Turbo | Gemini Pro | PaLM-2 | Claude-3 | Mixtral-8x7B | LLaMa-2-70B | LLaVa-1.5-13B | Random Guess |
|---|---|---|---|---|---|---|---|---|---|---|
| Even/Odd Function | Image | 56.3 | – | 47.9 | – | 43.2 | – | – | 24.7 | 33.3 |
| | LaTeX | **77.1** | 36.7 | **49.0** | **35.2** | 73.7 | 36.2 | **33.3** | – | |
| | Code | 67.7 | **45.1** | 48.2 | **35.2** | 77.6 | **41.7** | 33.3 | – | |
| Convex/Concave Function | Image | 74.8 | – | 54.3 | – | 49.6 | – | – | 35.2 | 50.0 |
| | LaTeX | 65.6 | 85.9 | 85.2 | 63.7 | 91.4 | 67.2 | 67.6 | – | |
| | Code | **66.02** | **89.1** | **87.9** | **67.6** | **93.0** | **69.1** | **74.2** | – | |
| Count Breakpoints | Image | 5.9 | – | 1.2 | – | 31.7 | – | – | 50.0 | 50.0 |
| | LaTeX | 86.7 | **100.0** | **100.0** | **100.0** | 98.4 | **100.0** | 0.0 | – | |
| | Code | **96.1** | **100.0** | **100.0** | **100.0** | 90.2 | 99.6 | 0.0 | – | |

Table 2: Benchmark results on *Mathematics* problems. We report accuracy scores. For each model, the best score of each task is highlighted in **bold**.

## 2.1 Mathematics

Our first evaluation suite consists of problems in continuous mathematics. Foundation models are increasingly deployed as assistants for data science applications (Cheng et al., 2023), and they must perform competently on plot understanding to provide helpful insights. We evaluate this capability by prompting foundation models to extract mathematical properties from continuous functions.

**Formats.** We consider three input formats: **Image**, **Text with LaTeX**, and **Text with Code**. We first sample candidate functions and convert them to their LaTeXand sympy representations. Then, we use matplotlib to plot the functions. In practice, it can be difficult to exactly map these images back to their corresponding functions, so the authors manually filtered data to ensure that the properties to be tested are clearly visible in the images.

**Tasks.** We consider three tasks: classifying parity (*i.e. even/odd/neither*) functions, *convex/concave* functions, as well as counting the number of *breakpoints* in piecewise linear functions (either 2 or 3). These problems are designed to be visually trivial for humans to solve, but requires some mathematical maturity to answer analytically. We sample 128 samples for each class, culminating to 896 total samples. We refer readers to Appendix A.1 for additional details and Appendix B for sample prompts and responses.

## 2.2 Games

A critical area for enhancing the strategic capabilities of foundation models lies in understanding game strategies. We focus on two tasks in the game of chess: winner identification and chess puzzle solving. For both tasks, we use data from the *Lichess database*[3].

**Formats.** For our chess game dataset, we consider four representations: Graphical Board (**Image**), Algebraic Notation Layout (**ANL**), Portable Game Notation (**PGN**), and Forsyth-Edwards Notation (**FEN**). The Graphical Board is an image of the chessboard, whereas ANL, PGN, and FEN are all text-based representations of the game state. For additional details, please see Appendix A.3.

**Tasks.** We consider two tasks: *winner identification* from final board states and *chess puzzles*. In the winner identification task, the objective is to analyze a given representation of a chess game and determine the outcome: whether the game resulted in a win for White, a win for Black, or a draw. Our dataset for this task includes 257 games that lasted more than nine rounds in February 2024. In the chess puzzles task, a chess puzzle is given and the objective is to find the best first move. The dataset for this task comprises 200 puzzles.

---

[3]https://lichess.org

| | | GPT-4 Turbo | GPT-3.5 Turbo | Gemini Pro | PaLM-2 | Claude-3 | Mixtral-8x7B | LLaMa-2-70B | LLaVa-1.5-13B | Random Guess |
|---|---|---|---|---|---|---|---|---|---|---|
| **Winner Identification** | Image | 54.3 | – | 45.7 | – | 41.5 | – | – | 0.0 | |
| | ANL | 50.4 | 45.7 | 27.5 | 44.6 | 60.1 | 40.7 | 25.4 | – | 33.3 |
| | PGN | **85.7** | **52.3** | **69.8** | **65.1** | 66.3 | **70.5** | **53.7** | – | |
| | FEN | 50.4 | 31.4 | 8.1 | 8.9 | **74.4** | 7.4 | 12.4 | – | |
| **Chess Puzzles** | Image | 1.0 | – | 0.0 | – | 0.5 | – | – | 0.0 | |
| | ANL | 1.0 | 0.0 | 0.0 | 0.5 | 1.0 | 0.0 | 0.5 | – | ~2.9 |
| | PGN | 0.0 | 0.0 | 0.0 | 0.5 | 0.5 | 0.0 | 0.5 | – | |
| | FEN | **4.5** | 0.0 | **1.0** | 0.5 | **2.0** | 0.0 | 0.5 | – | |

Table 3: Benchmark results on chess *Games* problems. We report accuracy scores. For each model, the best score of each task is highlighted in **bold**.

## 2.3 Algorithms

To provide concrete planning advice, foundation models need to reason algorithmically. We focus on graph algorithms, which are relevant in many realistic scenarios. For example, if asked to purchase flight tickets from Los Angeles to Vienna, foundation models need to (1) enumerate all possible choices with possible connection stops in between, and (2) check the lowest ticket prices among all choices. Fundamentally, the first task is a *graph connectivity* problem and the second is a *weighted shortest path* problem.

In this section, we investigate three main graph algorithm problems with increasing complexity: graph connectivity, maximum flow, and graph isomorphism.

**Formats.** For graph algorithms, we consider three representations: 1. **Image**, where graphs are visualized with `networkx` package with random styles; 2. **Text with LaTeX**, where the *adjacency matrix* is chosen to be the mathematical representation of graphs; 3. **Text with story or description**, where the graph problems are described as a story-telling version of the scenario (*e.g.*, formulating the graph connectivity problem as determining whether it is possible to drive from one city to another).

**Tasks.** We consider three tasks. *Graph Connectivity* requires deciding whether two query nodes are connected in an undirected graph. *Maximum Flow* requires computing the maximum flow one can send from a given source node to a given sink node, within a weighed directed graph. Finally, *Graph Isomorphism* requires determining whether two provided graphs with the same number of nodes are isomorphic (i.e., whether there exists a bijection between nodes that preserves all edges). Each task contains 128 problems with details in Appendix A.2.

## 2.4 Science

**Formats.** We consider two input formats for science questions: **Image** and **Text**. For the *Image* representation, each sample includes a textual question and multiple choices, along with a figure that provides additional context. To get the corresponding isomorphic *Text* inputs, one author manually wrote descriptions for each figure.[4] The annotator avoided introducing any extra reasoning or information beyond what is depicted in the figures. Isomorphic examples of image and text representation are provided in Appendix Table 7.

**Tasks.** We compiled a dataset consisting of 75 *Chemistry* and 75 *Physics* questions. We first selected questions from the ScienceQA (Lu et al., 2022) test set, choosing only questions

---

[4]In preliminary experiments, we leveraged GPT-4 to generate image descriptions. However, we observed that these descriptions often include reasoning and additional knowledge, leading to biased "improved" performance.

| | | GPT-4 Turbo | GPT-3.5 Turbo | Gemini Pro | PaLM-2 | Claude 3 | Mixtral-8x7B | LLaMa-2-70B | LLaVa-1.5-13B | Random Guess |
|---|---|---|---|---|---|---|---|---|---|---|
| Graph Connectivity | Image | 75.8 | – | 50.8 | – | 53.1 | – | – | 50.0 | |
| | Math | 80.5 | 53.1 | 50.0 | 53.9 | 82.0 | 62.5 | 50.0 | – | *50.0* |
| | Text | **95.1** | **78.1** | **75.0** | **85.9** | 94.5 | **84.4** | 50.0 | – | |
| Maximum Flow | Image | 36.7 | – | 13.3 | – | 12.5 | – | – | 2.3 | |
| | Math | 32.8 | **25.8** | 15.6 | 14.1 | 56.3 | 8.6 | **18.0** | – | *~4.0* |
| | Text | **56.3** | 20.3 | **19.5** | **45.3** | 73.4 | **14.1** | 6.3 | – | |
| Graph Isomorphism | Image | 50.0 | – | 46.9 | – | 51.6 | – | – | 50.0 | |
| | Math | **62.5** | 49.2 | **47.7** | **50.8** | 50.0 | 50.0 | 50.0 | – | *50.0* |
| | Text | 57.0 | **50.0** | 36.7 | 50.0 | 53.1 | 50.0 | 50.0 | – | |

Table 4: Benchmark results on graph *Algorithms* problems. For connectivity and isomorphism, binary classification accuracy is reported. For Maxflow, an exact match accuracy of total flows is reported. For each model, the best score of each task is highlighted in **bold**.

| | | GPT-4 Turbo | GPT-3.5 Turbo | Gemini Pro | PaLM-2 | Claude 3 | Mixtral-8x7B | LLaMa-2-70B | LLaVa-1.5-13B | Random Guess |
|---|---|---|---|---|---|---|---|---|---|---|
| Physics | Image | 74.7 | – | 61.3 | – | 70.7 | – | – | 42.7 | 38.3 |
| | Text | **86.7** | 84.0 | **69.3** | 76.0 | **89.3** | 68.0 | 61.3 | – | |
| Chemistry | Image | 69.3 | – | 72.0 | – | 72.0 | – | – | 45.3 | 38.3 |
| | Text | **94.7** | 76.0 | **85.3** | 84.0 | **90.7** | 65.3 | 73.3 | – | |

Table 5: Benchmark results on *Science* problems. We report accuracy scores. For each model, the best score of each task is highlighted in **bold**.

whose answers rely on understanding the corresponding figures. This process resulted in a total of 50 chemistry and 60 physics questions. To enhance the diversity of the science dataset, we manually added 25 new chemistry questions and 15 new physics questions, introducing three new categories distinct from those in the ScienceQA dataset. The category distribution of our entire science dataset is detailed in Appendix Figure 4, illustrating the wide range of science questions covered.

## 3 Performance Analysis

We evaluate IsoBench on API-access multimodal foundation models, such as GPT-4V (`gpt-4-0125-preview` and `gpt-4-vision-preview`), Claude 3 (`claude-3-opus-20240229`), and Gemini Pro (`gemini-1.0-pro` and `gemini-pro-vision`). We also benchmark on API-access single-modal large language models for reference, such as GPT-3.5 Turbo (OpenAI, 2023a, `gpt-3.5-turbo-0125`) and PaLM-2 (Anil et al., 2023, `text-bison-001`). We evaluted representative open-source models such as Mixtral 8x7B (Jiang et al., 2024), LLaMa-2 70B (Touvron et al., 2023), and LLaVA-1.5 7B (Liu et al., 2023a) as well. All models we evaluate are instruction-tuned models and are zero-shot prompted. We present fine-grained results on these tasks in Tables 2 to 5, and provide sample responses in Figures 5 to 17.

**Overview.** We observe that, across all models that admit multimodal representations, language-only prompts almost always outperform vision-language prompts. On language prompts, API-access models exhibit strong reasoning capabilities (*e.g.* Figure 13). They generally outperform open-sourced models, which exhibit a tendency to generate a *default*

*answer* regardless of input samples. We now delineate several intriguing observations. We present aggregated IsoBench results in Table 1 with details on more models in Table 8.

**Vision models may be insufficient.**    In tasks that require explicit enumeration of objects such as breakpoint and chemistry (*e.g.* Figures 12 and 17), we observe significant performance degradation due to counting errors, which has been a long-standing problem for vision models (Liu et al., 2022). In fact, leveraging a sub-optimal feature extractor for one modality may hurt the performance of a multimodal model; and the design of an optimal model requires nuanced understanding of the interaction effects between modalities (Liang et al., 2024). Thomason et al. (2018) also showed unimodal models can even outperform their multimodal counterparts, and it suggests that models can often learn a shortcut only relying on language model part and perform weaker when forced to use only images. Overall, visual recognition errors, such as Figures 6, 8 and 12, are prevalent in failure cases, and often misdirect natural language generation towards incorrect conclusions.

**Models cannot utilize low-level visual features for generation.**    Perhaps surprisingly, vision-language foundation models are far from perfect for convexity problems, which mostly feature simple, smooth curves, even though curve detectors (Olah et al., 2020) are among the most observed and well-studied phenomena in the mechanistic understanding of vision models. This inconsistency suggests that, even if performing certain tasks is within a vision model's capabilities, the current multimodal fusion scheme is insufficient for eliciting them. We posit that popular approaches such as image-tokenization only offer a coarse-grained representation that summarizes high-level features, and may be unsuitable for detailed analysis such as plot and chart understanding.

**LLM backbones may not be exempt from blame.**    Recent works (Gonen et al., 2022; Razeghi et al., 2022; Han & Tsvetkov, 2022) have reported language models to prefer textual formats that are more common in the pre-training data, which echo our observation in performance variability between different language-only inputs. While formats do not fully explain this discrepancy, we hypothesize that the imbalance between visual and input data, and in general the sub-optimal robustness of LLMs in the face of less common distribution can contribute to the performance gap between visual and language inputs.

Lastly, we observe generated responses of vision-language prompts to be cursory compared to those of language-only prompts; and they sometimes contain a direct answer instead of a reasoning process, despite being instructed to perform chain-of-thought reasoning. To mitigate this difference, we discuss two simple strategies that aims to augment a vision-language prompt with better reasoning capabilities of language prompts, presented next.

## 4   IsoCombination and IsoScratchPad

In many scientific fields, the images are complicated, researchers usually provide extra text inputs to help the model. This is also a common practice in scientific fields, where figures and tables in most papers are coupled with captions (for example, this paper), providing descriptions and takeaways. Motivated by these and based on the observations made above on IsoBench, we can design two simple and deliberately contrived mechanisms to further study the performance of multimodal foundation models on tasks with isomorphic representations: *IsoCombination* (IsoCB) and *IsoScratchPad* (IsoSP). IsoCB feeds multiple isomorphic representations to the multimodal model simultaneously in order to see the combined effect of using multiple representations on performance. Alternatively, IsoSP uses a scratchpad to explicitly translate from visual to text representations (and then uses this text representation to solve a task), taking advantage of the empirically higher-performing text representations for visual data. In particular, for IsoSP we employ a two-step prompting strategy: initially, the model receives a prompt featuring a visual representation, which it is tasked to translate into a textual format; then, the model is prompted with this generated text representation to predict the output. We illustrate both methods in Figure 3.

We show results for IsoCB and IsoSP in Table 6, along with results for image-only and text-only representations. Interestingly, we find that IsoSP, which converts from image to

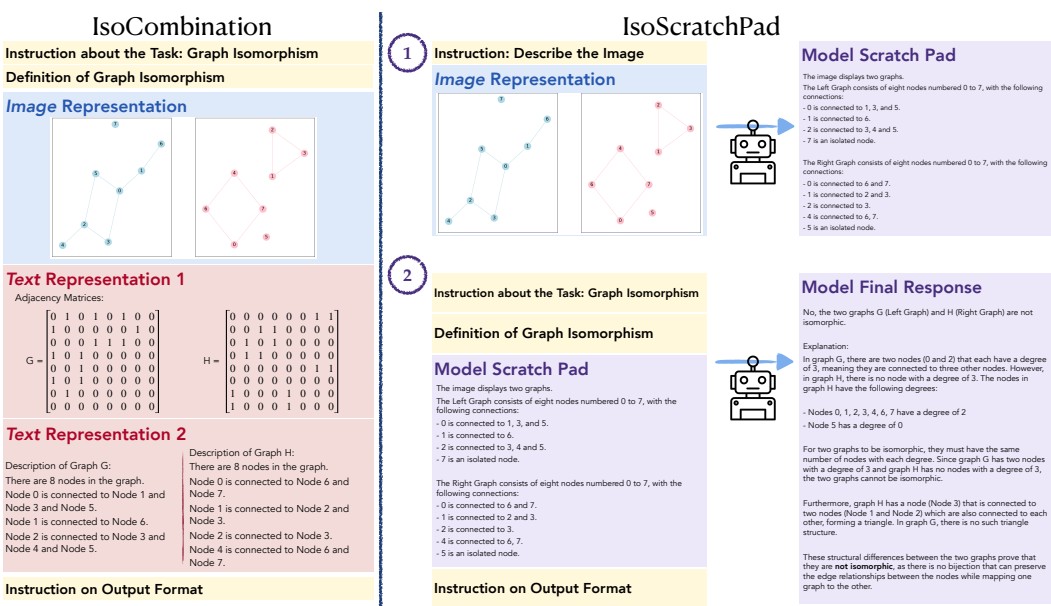

Figure 3: Illustration of IsoCombination (IsoCB) and IsoScratchPad (IsoSP). IsoCB combines all representations provided by a user and constructs one unified prompt for a foundation model. IsoSP is a two-step prompting method, where a foundation model first describes an image and then uses the textual description as the sole representation for a given task.

| Task | | GPT-4 Turbo | Gemini 1.0 Pro | Claude-3 Opus |
|---|---|---|---|---|
| Maxflow | Image | 36.7 | 13.3 | 12.5 |
| | Text (Best) | 56.3 (+19.5) | 19.5 (+6.3) | 73.4 (+60.9) |
| | IsoCB | **65.6** (+28.9) | **21.9** (+8.6) | **75.0** (+62.5) |
| | IsoSP | 52.3 (+15.6) | 19.5 (+6.3) | 22.7 (+10.2) |
| Connectivity | Image | 75.8 | 50.8 | 53.1 |
| | Text | **95.1** (+19.3) | **75.0** (+25.8) | **94.5** (+41.4) |
| | IsoCB | 83.6 (+7.8) | 52.3 (+1.6) | 85.2 (+32.0) |
| | IsoSP | 82.0 (+6.3) | 51.6 (+0.8) | 63.3 (+10.1) |
| Physics QA | Image | 74.7 | 61.3 | 70.7 |
| | Text | 86.7 (+12.0) | 69.3 (+8.0) | **89.3** (+18.7) |
| | IsoCB | **88.0** (+13.3) | 70.7 (+9.3) | 88.0 (+17.3) |
| | IsoSP | 84.0 (+9.3) | **76.0** (+14.7) | 78.7 (+8.0) |
| Chemistry QA | Image | 69.3 | 72.0 | 72.0 |
| | Text | **94.7** (+25.3) | **85.3** (+13.3) | **90.7** (+18.8) |
| | IsoCB | 92.0 (+22.7) | 82.7 (+20.7) | 89.3 (+27.3) |
| | IsoSP | 88.0 (+18.7) | 84.0 (+12.0) | 73.3 (+1.3) |

Table 6: IsoCombination and IsoScratchPad results. Best methods are highlighted in **red** and improvements over image-only prompts are in (green). We find that both methods improve performance in comparison with image representations, and for certain domains, IsoCombination additionally improves performance relative to text representations.

text, typically outperforms image-only representations (and thus it could potentially be used to improve performance given only image representations for some tasks). However, compared to direct text representations, IsoSP tends to fall short. Our manual review of IsoSP and text results provides a possible explanation: the model has difficulty understanding images, especially in tasks that require counting items or comparing numbers visually. The larger the performance gap between IsoSP and Text representations, the harder it becomes for the model to understand images. Another possible reason for IsoSP's underperformance relative to Text representations could be that, the model-generated interpretation of images tends to miss critical information for reasoning. For example, to determine which pair of

magnets in the given image has a larger magnetic force in a physics question, it is essential to consider both the distance between the magnets and their sizes. Yet, the descriptions generated by models like Claude-3 recognize only the distance, neglecting the magnet sizes, and thus leading to an incorrect answer. Additionally, we find that IsoCB, which combines representations, typically outperforms image-only representations, and even text-only representations on certain tasks.

## 5    Related Works

**Vision-Language Foundation Models.**    There has been a plethora of recent developments in vision-language foundation models; they can be broadly categorized by their methods for representing visual modalities. A representative approach involves *tokenizing* visual inputs to be jointly trained with language inputs (Yu et al., 2023; Google, 2023; McKinzie et al., 2024; Chen et al., 2022; 2023; Team, 2024, *inter alia*). Another line of work processes continuous visual features by directly projecting them to the language embedding space via a learnable function (Liu et al., 2023b;a; 2024a; Bavishi et al., 2023). While the design choices of some API-access models (OpenAI, 2023b; Anthropic, 2024; Google, 2023; Reka, 2024) remain largely unknown, most performant models use *early fusion*, the practice of integrating visual and language features at the input level of an autoregressive model. At the core of these design choices is the hardness in representing visual features, which has been reported by several early studies (McKinzie et al., 2024) to be the key bottleneck towards performant vision-language foundation models. IsoBench aims to supplement prior works, which have mostly focused on analyzing modeling methods, with a data-centric view by measuring performance discrepancies incurred by multimodal representations.

**Multimodal Datasets.**    Many vision-language datasets have recently been curated to assess the capabilities of multimodal foundation models (Yue et al., 2023; Lu et al., 2024; Zhang et al., 2024a; Fan et al., 2024; Zhang et al., 2024b). They often consist of a large collection of problems with varying levels of difficulty and usage of the visual inputs. As a result, they are suitable for assessing VLMs as a whole, but cannot attribute these performances to capabilities of their LLM backbones and improvements from downstream vision-language training. While several contemporaneous works (Fan et al., 2024; Zhang et al., 2024a;b) introduce a language-only subset of their vision-language prompts, they focus primarily on assessing certain areas of foundation models, such as deductive reasoning and problems of beyond-polynomial complexity. IsoBench is the first of its kind that offers a holistic and fine-grained analysis of the performance gap induced by variations in input modalities. For instance, a contemporaneous work, MathVerse (Zhang et al., 2024a), studies redundant, but not necessarily equivalent input representations on math problems whilst IsoBench studies the impact of isomorphic representations on a broader set of tasks.

**Sensitivity of Language Model to Perturbations.**    Similar to what IsoBench observes on multimodal foundation models' sensitivity to input modalities, uni-modal language models also show sensitivity to input perturbation. For example, LLMs are shown to be sensitive to subtle changes in zero-shot and few-show settings (Sclar et al., 2023; Chang & Jia, 2023). Meanwhile, the sensitivity of Transformers, the backbone of foundation models, is also broadly studied (Bombari & Mondelli, 2024; Bhattamishra et al., 2023; Vasudeva et al., 2024; Kong et al., 2024). To address the sensitivity issues, methods including prompt design (Yoo et al., 2021; Mishra et al., 2021; Le et al., 2023; Liu et al., 2023c) and prompting-based training strategies (Jain et al., 2023; Guo et al., 2023) have been employed. IsoBench is one of the pioneers to study model sensitivity to input representations and modalities.

**Foundation Models for Algorithmic Reasoning**    Recent work has shown that Transformers, the backbone of multimodal foundation models, are capable of implementing various algorithms after fine-tuning on particular tasks (Khalighinejad et al., 2023; Garg et al., 2022; von Oswald et al., 2022; Hanna et al., 2023; Fu et al., 2023; Lee et al., 2023; Zhou et al., 2024) or with proper prompting schemes (Wei et al., 2023; Yao et al., 2023; Liu et al., 2024b). However, the extent to which foundation models are capable of solving problems involving complex reasoning, algorithms, or scientific awareness remains unclear.

## 6 Conclusion

In this paper, we developed ***IsoBench***, a broad-coverage benchmark for evaluating multi-modal foundation models on a varied set of quantitative tasks, where each example input is provided with multiple isomorphic representations. IsoBench includes four major subjects (mathematical functions, algorithmic problems, science questions, and chess games) in a dataset consisting of over 1,630 examples. Within each domain, we evaluated multiple input representations—one with visual input and others with domain-specific, isomorphic textual inputs—to facilitate multimodal performance assessments. We found that all tested models perform substantially better on text-only prompts than image-based prompts, in contrast with known human preferences for images over text. To bridge performance discrepancies between different input modalities, we also introduced IsoCombination and IsoScratchPad, two procedures that, respectively, fuse input modalities and transforms visual input to textual representation at inference time. In full, we hope that IsoBench can provide fine-grained feedback to diagnose performance gaps caused by the form of a given input representation, and help us better understand how the capabilities of multimodal foundation models change depending on the input modality that is provided.

## Acknowledgement

We would like to thank the USC NLP, Duke NLP Group, and an Open Philanthropy University Organizer Fellowship for providing computing resources. DF and RJ were supported by an Open Philanthropy research grant and a Google Research Scholar Award. DF would like to thank Vatsal Sharan and Qilin Ye for their discussions on graphs. OL would like to thank Xinyan Velocity Yu, Bill Yuchen Lin, and Corby Rosset for their helpful discussions.

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

# Appendix

## A  Dataset Generation

We detail procedures to curate the IsoBench dataset. Please refer to the supplementary materials for concrete samples and further implementation details.

### A.1  Mathematics

1. **Image**: the model is prompted with a (textual) problem statement and the image visualization of the function. We generate these plots with `matplotlib`.
2. **Text (LATEX)**: in addition to the problem statement, we provide LATEXdefinition of the function as textual inputs.
3. **Text (Code)**: we replace the LATEXdefinition of function in the previous representation with its sympy definition. All other parts of the prompt are kept identical.

**Even & Odd Function.**   We sample a dataset of rational functions – the quotients of two polynomials – and extract their even and odd parts as samples. Each prompt formulates a 3-way classification problem for the model to choose among even, odd, and neither. Each class consists of 128 samples.

**Convex & Concave Function.**   We sample convex (resp. concave) functions by sampling operations from $\{x^p, |x|, -\log x, \exp x\}$ (resp. the negative of these operations) as well as multiplicative weights. Each class consists of 128 samples; and we prompt the model with their appropriate domains to ensure correctness of either of the properties.

**Counting Breakpoints.**   We sample piecewise linear functions with 2 or 3 breakpoints, *i.e.* sudden changes in slope at intersections of linear functions. We manually audit to ensure that the changes in slope are visible. Each class consists of 128 samples.

Images for math problems are generated with 300 DPI in JPEG format.

### A.2  Algorithms

1. **Image**: the model is prompted with an instruction of the problem to solve and the image visualization of the graph. For graphs, we use `networkx` to visualize.
2. **Text with LATEX**: the model is prompted with an instruction of the problem to solve and the mathematical expression of the graphs, and here we choose to use the *adjacency matrix* as the mathematical representation.
3. **Text with story or description**: the model is prompted with an instruction of the problem to solve and the story-telling description of the scenario. For example, formulate the graph connectivity problem as the possibility of driving from one city to another.

**Graph Connectivity.**   We sample 128 undirected graphs using Erdos-Renyi random graph generation method. Each pair of nodes has a probability of $p$ to be connected. The query nodes are also sampled at random and we balance the benchmark to have 50% of the sample having query nodes connected and the rest disconnected.

**Maximum Flow.**   We sample 128 weighted directed graphs. Each edge has a random integer flow uniformly sampled from 0 to 9 and a fairly sampled direction. Maxflow is a fairly hard problem so we restrict the graphs to have 3, 4 or 5 nodes.

**Graph Isomorphism.**   We sample 128 pairs of graphs, with 64 pairs isomorphic and 64 of them not isomorphic. In generating isomorphic pairs $G$ and $H$, we sample a random permutation matrix $\Pi$, and the adjacency matrix corresponding to graph $H$ is $A_H = \Pi A_G \Pi^\top$ where $A_G$ is the adjacency matrix of graph $G$. In generating non-isomorphic pairs, $A_G$ and

$A_H$ are sampled individually. To avoid models using simple shortcuts such as counting the number of nodes, graphs $G$ and $H$ in each pair are guaranteed to have the same number of nodes and the same number of edges.

Images for algorithm problems are generated with 300 DPI in JPEG format.

### A.3 Games

1. **Graphical Board**: the model is prompted with an instruction of the problem to solve and the visual representation of the chess board and its pieces in a graphical format (PNG) is given.

2. **Algebraic Notation Layout (ANL)**: this refers to a text-based representation of the current position of pieces on the chess board, using algebraic notation for the squares.

3. **Portable Game Notation (PGN)**: this is a textual representation of the chess game's moves in standard algebraic notation. PGN is widely used for the recording of games, allowing for both humans and computers to read the game history. It captures the entire sequence of moves from the beginning to the current state or the end of the game.

4. **Forsyth-Edwards Notation (FEN)**: this is a compact way to describe the current position of a game of chess in textual format.

**Winner Identification from Checkmate**  The task is to analyze a given chess game representation and identify the winner: whether the game resulted in a win for White or Black, or if it ended without a winner. Our dataset for this task contains the games that lasted more than nine rounds in February 2024.

**Chess Puzzles**  For this task, we consider chess puzzles that consist of a chess position, along with a sequence of optimal moves necessary to resolve the puzzle. In our evaluation, the primary objective is to predict the initial move.

Images for Chess are the same as Lichess, and are converted to JPEG formats.

### A.4 Sciences

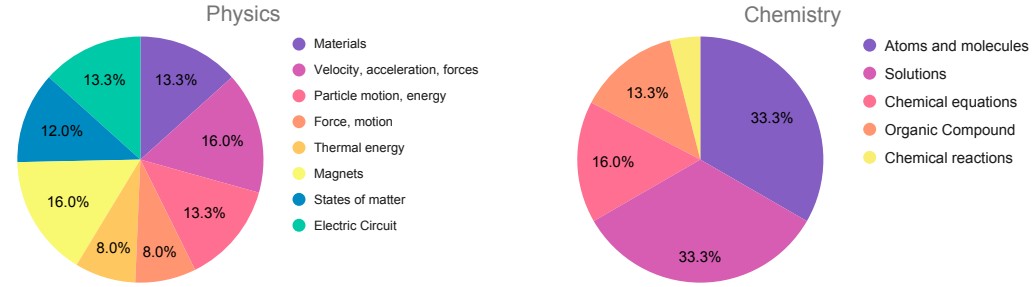

Figure 4: Distribution of science problem categories. We introduced three new categories that are absent from the ScienceQA dataset: Electric Circuit, Organic Compound, and Chemical equations.

1. **Image**: Each sample contains the textual questions, choices, and an accompanying figure providing additional context to the question.

2. **Text**: Instead of prompting LLMs to describe the image content, we asked one author to manually write descriptions for each figure. The annotator was instructed to avoid including additional reasoning or information beyond what is present in the figure, focusing solely on describing the content.

Images for Chess are the same as ScienceQA, and are converted to JPEG formats.

## B Sample Responses for *Math*, *Algorithms*, *Games*, and *Science* Problems

We show sample responses for *Math*, *Algorithms*, *Games*, and *Science* Problems in the figures below, for multiple multimodal foundation models.

---

**PROMPT**

You are given a real-valued, scalar function $f(x)$.
YOUR TASK is to determine whether $f(x)$ is an even function, an odd function, or neither.

- Definition of an even function: A function such that

$$f(x) = f(-x)$$

  where the value remains unchanged if the sign of the independent variable is reversed.
- Definition of an odd function: A function such that

$$f(-x) = -f(x)$$

  where the sign is reversed but the absolute value remains the same if the sign of the independent variable is reversed.
- A function is neither even nor odd if it does not satisfy either condition.

Here is the expression of $f(x)$:

$$f(x) = \frac{7x^6 - 3x^5 + x^2 - 21.76}{2x^6 + 4x^4 + 3x^2 + 11.34}$$

Respond with 'even', 'odd', 'neither' first on whether the function f(x) is even, odd, or neither, based on the definitions and your observation of the function.
Please parse it to JSON format. Put the result (even, odd, or neither) under key 'answer', and put the explanations and reasonings under key 'explanation' (Do not summarize, paraphrase or change meanings).

---

**GPT-4 RESPONSE**

to determine whether the function

$$f(x) = \frac{7x^6 - 3x^5 + x^2 - 21.76}{2x^6 + 4x^4 + 3x^2 + 11.34}$$

is even, odd, or neither, we substitute $-x$ for $x$ and compare the resulting expression to the original function. for an even function, we should find that

$$f(x) = f(-x)$$

and for an odd function, we should find that

$$f(-x) = -f(x)$$

substituting $-x$ into the function, we get

$$f(-x) = \frac{7(-x)^6 - 3(-x)^5 + (-x)^2 - 21.76}{2(-x)^6 + 4(-x)^4 + 3(-x)^2 + 11.34} = \frac{7x^6 + 3x^5 + x^2 - 21.76}{2x^6 + 4x^4 + 3x^2 + 11.34}$$

this expression is not equal to the original function $f(x)$, indicating that the function is not even. additionally, the expression is not the negative of the original function, indicating that the function is not odd. the presence of the $x^5$ term, which changes sign when $x$ is replaced with $-x$, but does not satisfy the condition for an odd function due to the other terms, is a key factor in determining that the function is neither even nor odd.

---

Figure 5: Sample response from GPT-4 to a parity problem in IsoBench. GPT-4 was able to analyze the parity of the rational function with the correct reasoning and computation.

**VISUALIZATION**

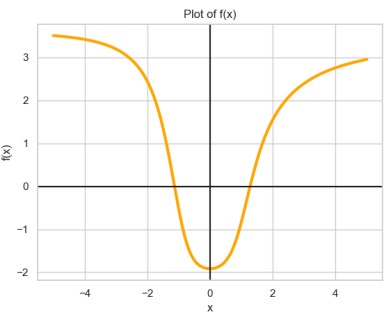

**PROMPT**
You are given a plot of a real-valued, scalar function f(x).
YOUR TASK is to determine whether f(x) is an even function, an odd function, or neither.

- Definition of an even function: A function such that

$$f(x) = f(-x)$$

  where the value remains unchanged if the sign of the independent variable is reversed.

- Definition of an odd function: A function such that

$$f(-x) = -f(x)$$

  where the sign is reversed but the absolute value remains the same if the sign of the independent variable is reversed

- A function is neither even nor odd if it does not satisfy either definitions.

Respond with 'even', 'odd', 'neither' first on whether the function f(x) is even, odd, or neither, based on the definition above. For any of your conclusions (even, odd, or neither), please explain in details based on your observation of the plot. Answer (start with 'even', 'odd', or 'neither'):

**GPT-4 RESPONSE**
The plot of the function $f(x)$ shows symmetry about the $y$-axis. This means that for every point $(x, f(x))$ on the right side of the $y$-axis, there is a corresponding point $(-x, f(x))$ on the left side of the y-axis with the same function value. This symmetry indicates that
$$f(x) = f(-x)$$
for all x in the domain of the function, which is the definition of an even function. Therefore, based on the plot, $f(x)$ is an even function.

Figure 6: Sample response from GPT-4V to the same parity problem as in Figure 5. GPT-4V cannot distinguish the nuanced asymmetry at the top-rigth and top-left corners of the plot. We observe the same failure mode for Gemini-Pro and LLaVA.

**PROMPT**
You are given a real-valued, scalar function $f(x)$.
YOUR TASK is to determine whether $f(x)$ is an even function, an odd function, or neither.

- Definition of an even function: A function such that

$$f(x) = f(-x)$$

where the value remains unchanged if the sign of the independent variable is reversed.

- Definition of an odd function: A function such that

$$f(-x) = -f(x)$$

where the sign is reversed but the absolute value remains the same if the sign of the independent variable is reversed.

- A function is neither even nor odd if it does not satisfy either condition.

Here is the expression of $f(x)$:

$$f(x) = \frac{4x^3 \left(x^2 - 1\right)}{16x^{10} + 8x^8 + 10x^2 + 13.98}$$

Respond with 'even', 'odd', 'neither' first on whether the function f(x) is even, odd, or neither, based on the definitions and your observation of the function.
Please parse it to JSON format. Put the result (even, odd, or neither) under key 'answer', and put the explanations and reasonings under key 'explanation' (Do not summarize, paraphrase or change meanings).

**GPT-4 RESPONSE**
to determine whether the function

$$f(x) = \frac{4x^3 \left(x^2 - 1\right)}{16x^{10} + 8x^8 + 10x^2 + 13.98}$$

is even, odd, or neither, we substitute $-x$ for $x$ and compare the result to the original function. substituting $-x$ into the function gives

$$f(-x) = \frac{4(-x)^3 \left((-x)^2 - 1\right)}{16(-x)^{10} + 8(-x)^8 + 10(-x)^2 + 13.98} = \frac{-4x^3 \left(x^2 - 1\right)}{16x^{10} + 8x^8 + 10x^2 + 13.98}$$

, which simplifies to $-f(x)$. this means that the function satisfies the condition for being an odd function, which is

$$f(-x) = -f(x)$$

. the function does not satisfy the condition for being an even function, which would require

$$f(x) = f(-x)$$

. therefore, the function is odd.

Figure 7: Sample response from GPT-4 to a parity problem in IsoBench. GPT-4 was able to analyze the parity of the rational function with the correct reasoning and computation.

**VISUALIZATION**

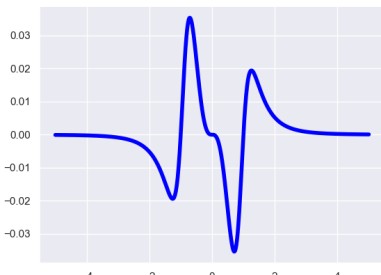

**PROMPT**
You are given a plot of a real-valued, scalar function f(x).
YOUR TASK is to determine whether f(x) is an even function, an odd function, or neither.

- Definition of an even function: A function such that

$$f(x) = f(-x)$$

  where the value remains unchanged if the sign of the independent variable is reversed.

- Definition of an odd function: A function such that

$$f(-x) = -f(x)$$

  where the sign is reversed but the absolute value remains the same if the sign of the independent variable is reversed

- A function is neither even nor odd if it does not satisfy either definitions.

Respond with 'even', 'odd', 'neither' first on whether the function f(x) is even, odd, or neither, based on the definition above. For any of your conclusions (even, odd, or neither), please explain in details based on your observation of the plot. Answer (start with 'even', 'odd', or 'neither'):

**GPT-4 RESPONSE**
neither
The function $f(x)$ depicted in the plot does not satisfy the conditions for being an even function nor an odd function. For a function to be even, the graph of the function should be symmetric with respect to the y-axis, meaning that the part of the graph in the positive $x$-region should be a mirror image of the part in the negative $x$-region. This function does not exhibit such symmetry; the shapes and values of the function on either side of the y-axis are not mirror images of each other.
Similarly, for a function to be odd, the graph should have rotational symmetry about the origin, which means that rotating the graph 180 degrees around the origin should result in the same graph. This is not the case for the given function either, as the positive and negative sides of the graph do not match when one side is rotated 180 degrees.
Since the function does not meet the criteria for being even or odd, we conclude that the function is neither even nor odd.

Figure 8: Sample response from GPT-4V to the same parity problem as in Figure 7. GPT-4V cannot reason with the symmetry at origin. We observe the same failure mode for all other vision models.

---

**PROMPT**

You are given a real-valued, scalar function f(x).
YOUR TASK is to determine whether f(x) is an convex function or an concave function

- Definition of a convex function: A function such that for all $x$, $y$, and 0 <= t <= 1

$$f(tx + (1-t)y) \leq tf(x) + (1-t)f(y)$$

- Definition of a concave function: A function such that for all $x$, $y$, and 0 <= t <= 1

$$f(tx + (1-t)y) \geq tf(x) + (1-t)f(y)$$

Here is the expression of $f(x)$, defined for all $x > 0$:

$$f(x) = 0.31x - 1.73 \log(x) + 1.13 |x| - 0.63$$

Respond with 'convex' or 'concave' first on whether the function $f(x)$ is convex or concave, based on the definitions and your observation of the function.
Please parse it to JSON format. Put the result (convex or concave) under key 'answer', and put the explanations and reasonings under key 'explanation' (Do not summarize, paraphrase or change meanings).

**Claude 3 RESPONSE**

The given function $f(x)$ is a convex function. This can be determined by examining the individual terms in the expression:

1. The term $0.31x$ is a linear function, which is both convex and concave.

2. The term $-1.73 \log(x)$ is a concave function, as the logarithm function is concave for $x > 0$.

3. The term $1.13|x|$ is a convex function, as the absolute value function is convex.

4. The constant term -0.63 does not affect the convexity or concavity of the function.

Since the sum of a convex function and a concave function is not guaranteed to be either convex or concave, we need to consider the dominant term. In this case, the convex term $1.13|x|$ grows faster than the concave term $-1.73 \log(x)$ as x increases. Therefore, for sufficiently large values of x, the convex term will dominate, making the overall function f(x) convex.

Figure 9: Sample response from Claude 3 to a convexity problem in IsoBench. Claude 3 was able to analyze the convexity of the function with the correct reasoning and computation.

---

**VISUALIZATION**

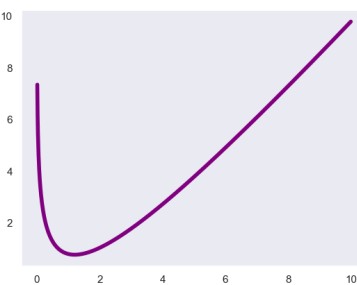

**PROMPT**
You are given a plot of a real-valued, scalar function f(x).
YOUR TASK is to determine whether $f(x)$ is an convex function or a concave function

- Definition of a convex function: A function such that for all $x$, $y$, and 0 <= t <= 1

$$f(tx + (1-t)y) \leq tf(x) + (1-t)f(y)$$

- Definition of a concave function: A function such that for all $x$, $y$, and 0 <= t <= 1

$$f(tx + (1-t)y) \geq tf(x) + (1-t)f(y)$$

Respond with 'convex', 'concave' first on whether the function $f(x)$ is convex or concave based on the definition above. For any of your conclusions (convex or concave), please explain in details based on your observation of the plot.
Answer (start with 'convex' or 'concave'):

**Claude 3 RESPONSE**
concave
Based on the plot, the function $f(x)$ appears to be concave. The key observations are:

1. The curve is continuously increasing, but the rate of increase is slowing down as x gets larger. This can be seen from the curve "flattening out" and having a decreasing slope as x increases.

2. If you draw a line segment connecting any two points on the curve, that line segment lies entirely below the curve. This is a key property of concave functions.

3. Intuitively, a concave function curves inward like the interior of a circle or sphere. The plot shows this inward curving shape characteristic of concave functions.

4. In contrast, a convex function would curve outward with an increasing slope, and line segments between points would lie above the curve.

So in summary, the decreasing slope, the inward curving shape, and the fact that line segments between points lie below the curve all clearly indicate that the plotted function f(x) is concave, not convex.

---

Figure 10: Sample response from Claude 3 to the same convexity problem as in Figure 9.
Claude 3 cannot distinguish the orientation of the curve at all.

**PROMPT**
You are given a real-valued, scalar function $f(x)$.
YOUR TASK is to count the number of breakpoints in the plot of $f(x)$. A breakpoint refers to a point on the function's domain at which the function changes its slope.
Here is the expression of $f(x)$:

$$f(x) = \begin{cases} 28.88x + 375.32 & \text{for } -14.9 < x \le -9.53 \\ -18.18x - 73.3 & \text{for } -9.53 < x \le 1.47 \\ 54.18x - 179.55 & \text{for } 1.47 < x \le 5.16 \\ 277.33 - 34.43x & \text{for } 5.16 < x \le 10.95 \end{cases}$$

Respond with the number of breakpoints (in Arab digits) first on how many breakpoints the function f(x) contains based on the definition and your observation of the function. You should IGNORE the left and right end point of the domain, i.e. if the function is defined on [a, b], you should only consider the domain (a, b).
Please parse it to JSON format. Put the result (in Arab digits) under key 'answer', and put the explanations and reasonings under key 'explanation' (Do not summarize, paraphrase or change meanings).

**Claude 3 RESPONSE**
The function $f(x)$ is a piecewise function defined over 4 intervals. The endpoints of these intervals are -14.9, -9.53, 1.47, 5.16, and 10.95. Ignoring the leftmost and rightmost endpoints as per the instructions, we consider the points -9.53, 1.47, and 5.16. At each of these points, the function changes its slope (i.e., the coefficient of $x$ changes). Therefore, these three points are breakpoints of the function $f(x)$. Thus, the function $f(x)$ has 3 breakpoints within its domain, excluding the endpoints.

Figure 11: Sample response from Claude 3 to a breakpoint problem in IsoBench. Claude 3 was able to identify the exact locations of the breakpoints. We observe similar behaviors for all API-access foundation models.

**VISUALIZATION**

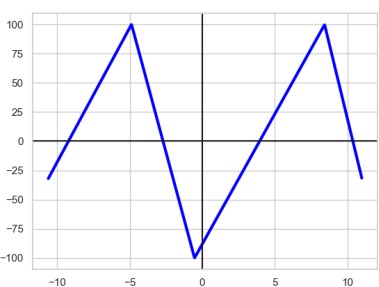

**PROMPT**
You are given a plot of a real-valued, scalar function $f(x)$.
YOUR TASK is to count the number of breakpoints in the plot of $f(x)$. A breakpoint refers to a point on the function's domain at which the function changes its slope. You should IGNORE the left and right end point of the domain, i.e. if the function is defined on $[a,b]$, you should only consider the domain $(a,b)$.
Respond with the number of breakpoints (in Arab digits) first on how many breakpoints the function $f(x)$ contains based on the definition above. For any of your result, please explain in details based on your observation of the plot.
Answer (in Arab digits):

**Claude 3 RESPONSE**
Answer (in Arab digits): 2
Explanation: Based on the plot of the function $f(x)$, I observe two breakpoints within the interior of the domain, excluding the endpoints.
The first breakpoint occurs at $x = 0$. As $x$ approaches 0 from the negative side, the function has a positive slope. However, as $x$ crosses 0 and becomes positive, the function changes to a negative slope. This change in slope direction at $x = 0$ indicates a breakpoint.
The second breakpoint is located at approximately $x = 5$. From $x = 0$ to $x = 5$, the function has a negative slope. But as x exceeds 5, the function's slope becomes positive again. This transition from a negative to a positive slope around $x = 5$ signifies another breakpoint.
The left and right endpoints of the function's domain are not considered breakpoints according to the given definition. Therefore, focusing only on the interior of the domain, the function $f(x)$ contains exactly 2 breakpoints based on the changes in slope observed in the provided plot.

Figure 12: Sample response from Claude 3 to the same breakpoint problem as in Figure 11. The behavior of Claude 3 is a combination of failing to account for all the breakpoints and/or reason with respect to the boundary points. We observe similar behaviors for all multimodal foundation models.

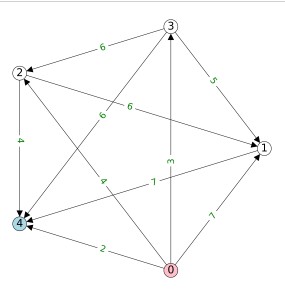

**VISUALIZATION**

**PROMPT**

You are given an adjacency matrix of a graph and two query nodes. (one source node and one sink node). The source node is the node where the flow starts and the sink node is the node where the flow ends.

YOUR TASK is to solve the maxflow problem given the weighted directed graph. Definition of Maxflow problem:

In the max flow problem, we have a directed graph with a source node s and a sink node t, and each edge has a capacity (integer valued, colored in green) that represents the maximum amount of flow that can be sent through it.

The goal is to find the maximum amount of flow that can be sent from s to t, while respecting the capacity constraints on the edges.

Query Example:

adjacency matrix:

[0 7 4 3 2]
[0 0 0 0 7]
[0 6 0 0 4]
[0 5 6 0 6]
[0 0 0 0 0]

Source node (zero-indexed): 0

Sink node (zero-indexed): 4

In the query example, the nodes are zero-indexed.

Compute the maximum flow from the source node to the sink node.

**CLAUDE-3 RESPONSE**

To solve the maximum flow problem using the given adjacency matrix, we can use the Ford-Fulkerson algorithm with the Edmonds-Karp implementation. The algorithm finds the maximum flow by repeatedly finding augmenting paths from the source to the sink and updating the residual graph until no more augmenting paths can be found.

Given:

- Adjacency matrix:

[0 7 4 3 2]
[0 0 0 0 7]
[0 6 0 0 4]
[0 5 6 0 6]
[0 0 0 0 0]

- Source node: 0
- Sink node: 4

Step 1: Initialize the flow matrix with all values set to 0.

Step 2: Find an augmenting path from the source to the sink using BFS.

- Augmenting path 1: 0 -> 1 -> 4

Residual capacity: min(7, 7) = 7

Update the flow matrix and residual graph.

Step 3: Find another augmenting path.

- Augmenting path 2: 0 -> 2 -> 4

Residual capacity: min(4, 4) = 4

Update the flow matrix and residual graph.

Step 4: Find another augmenting path.

- Augmenting path 3: 0 -> 3 -> 4

Residual capacity: min(3, 6) = 3

Update the flow matrix and residual graph.

Step 5: Find another augmenting path.

- Augmenting path 4: 0 -> 4

Residual capacity: 2

Update the flow matrix and residual graph.

Step 6: No more augmenting paths can be found. The maximum flow is the sum of the flows on the edges from the source to the sink.

Maximum flow = 7 + 4 + 3 + 2 = 16

Figure 13: Sample response from Claude-3-Opus to a maxflow problem in IsoBench.

---

**PROMPT**

You are given a FEN representation of a chess puzzle for which a sequence of unique best moves is determinable (e.g. sequences of moves leading to a forced checkmate).

- Definition of the Chess Puzzle: In a chess puzzle, you are required to make a series of optimal moves leading to checkmate, starting from the given position.

- YOUR TASK is to predict THE FIRST MOVE that should be played given this board setup.

- Your answer should first specify the move in Algebraic Coordinate Notation (e.g., "d2d1", "e5a1", "c4f4"). Afterward, provide your reasoning for why you chose that as your first move.

$$\textbf{FEN}: r2qk2r/ppp2ppp/2n2n2/6B1/1b2N1b1/5N2/PPP1PPPP/R2QKB1RwKQkq - 68$$

Answer (start with the first move in Algebraic Coordinate Notation like "e2e4" or "d7d5"):

---

**GPT-4 RESPONSE**

First Move: "a2a3"

Reasoning: In this position, given by the FEN "r2qk2r/ppp2ppp/2n6/6B1/1b2N1b1/5N2/PPP1PPPP/R2QKB1R w KQkq - 6 8", it is White's turn to move. The board setup reveals a significant tension in the center and around the kingside, particularly with both bishops (the Black bishop on g4 and the White bishop on g5) exerting pressure.

The immediate threat to address is the pin of the White knight on f3 by the Black bishop on g4. Moving the h3 pawn to attack the bishop would be tempting to relieve the pin on the knight; however, it weakens White's kingside structure and can be met by h5 from Black, securing the bishop's position.

A crucial observation here is the Black bishop on b4, which is currently pinning the White knight on e4, preventing it from moving without losing the queen. The first move, "a2a3", is aimed at challenging this bishop. This move forces the Black bishop to make a decision; it can either retreat or be captured if it remains on the b4 square. Retreating the bishop to a5 or c5 does not alleviate Black's positional difficulties significantly, and capturing on c3 would lead to bxc3, strengthening White's center and opening lines for the rook and bishop.

This move is selected over other aggressive or defensive options because it immediately addresses a piece that contributes to pinning and pressure without compromising White's position or losing material. By initiating this challenge, White aims to disrupt Black's coordination and gain a tempo to improve their position, possibly by preparing to castle, unpin the knight on f3, or increase control in the center. This move also preserves the option of castling kingside for White, keeping the king's safety as a priority.

---

Figure 14: Sample response from GPT-4 to a chess puzzle problem in IsoBench.

**VISUALIZATION**

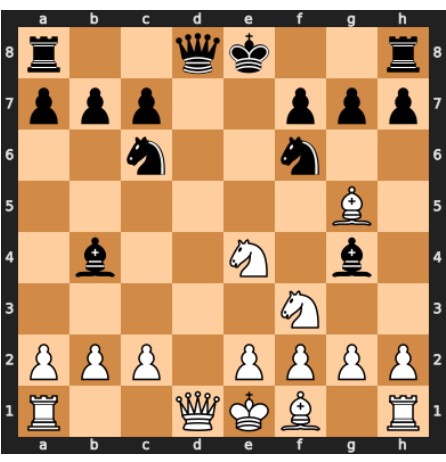

**PROMPT**

You are given a visual representation of a chess puzzle for which a sequence of unique best moves is determinable (e.g. sequences of moves leading to a forced checkmate).

- Definition of the Chess Puzzle: In a chess puzzle, you are required to make a series of optimal moves leading to checkmate, starting from the given position.
- YOUR TASK is to predict THE FIRST MOVE that should be played given this board setup.
- Your answer should first specify the move in Algebraic Coordinate Notation (e.g., "d2d1", "e5a1", "c4f4"). Afterward, provide your reasoning for why you chose that as your first move.

Answer (start with the first move in Algebraic Coordinate Notation like "e2e4" or "d7d5"):

**GPT-4 RESPONSE**

Answer: Bg5f6

The best move for white to play in this position is Bg5xf6. The reason behind this move is that by capturing the knight on f6 with the white bishop, white is achieving multiple objectives:

White is removing a defender of the d7 square, which could potentially be an entry point for the white queen or other pieces if the knight on c6 moves. The g7 pawn is pinned by the white bishop on b2, and cannot recapture on f6 without losing material since the rook on h8 would be left undefended. This move also opens up the e-file for the white queen and rook to become more active, possibly leading to an attack on the black king, who is still in the center of the board and unable to castle due to the king's move earlier. Thus, Bg5xf6 appears to be the strongest move in the position, setting up threats and potential for further penetration into black's position.

Figure 15: Sample response from GPT-4V to the same chess puzzle problem as in Figure 14.

```
PROMPT
You are given a multiple-choice chemistry question.
YOUR TASK is to read the question and select the correct answer from the provided options.

In Solution A and Solution B, the green particles represent the solute. The volume of the solvent in both containers are 20
mL. Solution A has three green particles, and Solution B has three green particles.

Which solution has a higher concentration of green particles?
A. Solution B
B. neither; their concentrations are the same
C. Solution A

GPT-4 RESPONSE
If Solution A and Solution B have the same number of green particles (solute) and the volume of the solvent in both containers
is equal, then their concentrations of green particles are the same. Therefore, the correct answer is:
B. neither; their concentrations are the same
```

Figure 16: Sample response from GPT-4 to a multiple-choice chemistry question in IsoBench.

```
VISUALIZATION
```

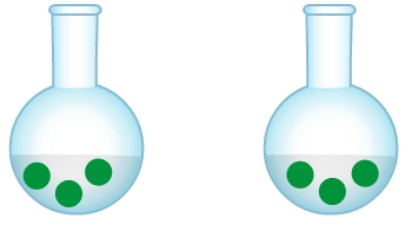

```
                                 Solvent volume: 20 mL      Solvent volume: 20 mL

                                        Solution A                 Solution B
PROMPT
You are given a multiple-choice chemistry question.
YOUR TASK is to read the question and select the correct answer from the provided options.

Which solution has a higher concentration of green particles?
A. Solution B
B. neither; their concentrations are the same
C. Solution A

GPT-4 RESPONSE
Solution A has a higher concentration of green particles. There are more green particles in Solution A than in Solution B,
and both solutions have the same volume of solvent (20 mL).
```

Figure 17: Sample response from GPT-4V to the same multiple-choice chemistry question as
in Figure 16.

| Image Representation | Text Representation |
|---|---|
| 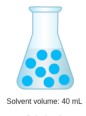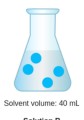
Solvent volume: 40 mL  Solvent volume: 40 mL
**Solution A**    **Solution B**

Which solution has a higher concentration of blue particles?

0. Solution B  1. Solution A  2. neither; their concentrations are the same | In Solution A and Solution B, the blue particles represent the solute. The volume of the solvent in both containers are 40mL. Solution A has 8 blue particles, and Solution B has 4 blue particles.

Which solution has a higher concentration of blue particles?

0. Solution B  1. Solution A  2. neither; their concentrations are the same |
| $2\text{Na}\ (s) + \text{H}_2\text{O}\ (l) \rightarrow 2\text{NaOH}\ (aq) + \text{H}_2\ (g)$

The figure shows a reaction between sodium (Na) and water (H2O). Which best explains why the equation is not balanced?

0. There is too much sodium in the reactants.

1. There is not enough water in the reactants.

2. There is too much hydrogen gas in the products.

3. There is not enough sodium hydroxide in the products. | This is a chemical equation representing the reaction between solid sodium (Na) and liquid water (H2O) to form aqueous sodium hydroxide (NaOH) and hydrogen gas (H2). In the reaction, two moles of sodium react with one mole of water to produce two moles of sodium hydroxide and one mole of hydrogen gas.

Which best explains why the equation is not balanced?

0. There is too much sodium in the reactants.

1. There is not enough water in the reactants.

2. There is too much hydrogen gas in the products.

3. There is not enough sodium hydroxide in the products. |
| 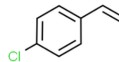

How many chlorine atoms are there in the given organic molecule?

0. 0   1. 1   2. 2   3. 3 | How many chlorine atoms are there in a given organic molecule? Its molecular formula is C8H7Cl.

0. 0   1. 1   2. 2   3. 3 |
| 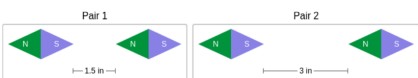

Think about the magnetic force between the magnets in each pair. Which of the following statements is true?

0. The magnitude of the magnetic force is the same in both pairs.

1. The magnitude of the magnetic force is greater in Pair 2.

2. The magnitude of the magnetic force is greater in Pair 1. | The distance between the magnets in Pair 1 is 1.5 in, while the distance between the magnets in Pair 2 is 3 in.

Think about the magnetic force between the magnets in each pair. Which of the following statements is true?

0. The magnitude of the magnetic force is the same in both pairs.

1. The magnitude of the magnetic force is greater in Pair 2.

2. The magnitude of the magnetic force is greater in Pair 1. |
| 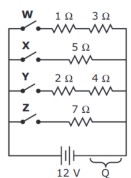   The diagram shows a circuit with four switches: W, X, Y, and Z. A student wants the largest possible current to pass through location Q when the student closes one of the switches. Which switch should the student close?
0. W    1. X     2. Y       3. Z | The circuit is a mixed electrical circuit consisting of both series and parallel connections. There are four main branches and each branch contains one of the four switches: W, X, Y, and Z. The branch with the W switch contains two resistors (1 ohm and 3 ohm) connected in series. The branch with the X switch contains a single resistor (5 ohm). The branch with the Y switch contains two resistors (2 ohm and 4 ohm) connected in series. The branch with the Z switch contains a single resistor (7 ohm).

A student wants the largest possible current to pass through location Q when the student closes one of the switches. Which switch should the student close?

0. W   1. X   2. Y   3. Z |

Table 7: Examples of isomorphic representations on *Science* problems.

## C   Full IsoBench Results

| | | GPT-4 Turbo | Gemini Pro | Claude 3 | GPT-3.5 Turbo | PaLM-2 | Mixtral-8x7B | LLaMa-2-70B | LLaVa-1.5-13B | Random Guess |
|---|---|---|---|---|---|---|---|---|---|---|
| Science | Image | 72.0 | 66.7 | 71.3 | – | – | – | – | 44.0 | *38.3* |
| | Text | 86.7 | 69.3 | 89.3 | 84.0 | 76.0 | 68.0 | 61.3 | – | |
| Mathematics | Image | 46.9 | 36.4 | 41.7 | – | – | – | – | 34.9 | *44.4* |
| | Text (LaTeX) | 76.6 | 73.9 | 85.8 | 68.9 | 61.8 | 63.3 | 33.6 | – | |
| | Text (SymPy) | 75.3 | 73.9 | 85.6 | 73.3 | 63.0 | 66.1 | 35.5 | – | |
| Algorithms | Image | 54.2 | 37.0 | 39.1 | – | – | – | – | 34.1 | *34.7* |
| | Text (LaTeX) | 58.6 | 37.8 | 62.8 | 42.7 | 42.2 | 40.4 | 39.3 | – | |
| | Text (Story) | 69.5 | 43.8 | 73.7 | 49.5 | 55.2 | 49.5 | 35.4 | – | |
| Games | Image | 27.6 | 22.9 | 21.0 | – | – | – | – | 0.00 | *18.1* |
| | Text (ANL) | 25.7 | 13.8 | 30.5 | 22.9 | 22.5 | 20.4 | 12.9 | – | |
| | Text (FEN) | 27.4 | 4.7 | 38.2 | 15.7 | 4.6 | 3.7 | 6.5 | – | |
| | Text (PGN) | 42.8 | 34.9 | 33.4 | 26.2 | 32.8 | 35.3 | 27.1 | – | |
| Average | Image | 50.2 | 40.7 | 43.3 | – | – | – | – | 28.3 | *33.9* |
| | Text (Best) | 68.9 | 55.6 | 71.9 | 58.2 | 56.8 | 57.2 | 39.8 | – | |
| | Gap | 18.7 | 14.9 | 28.7 | – | – | – | – | – | |

Table 8: IsoBench results with all API-access models and open-source models. All tasks of the four major domains are included.

