# OpenReview forum: "IsoBench: Benchmarking Multimodal Foundation Models on Isomorphic Representations"
_colmweb.org/COLM/2024/Conference — COLM_

### Official Review · Reviewer_R3GK · 2024-04-15

**Rating:** 7
**Confidence:** 3
**Ethics Flag:** 1

**Summary:**

The contribution outlined in the passage is centered around exploring the impact of input modality on the capabilities of current foundation models. The authors introduce IsoBench, a benchmark dataset comprising problems from various domains like math, science, algorithms, and games. Each problem in IsoBench is presented with multiple representations, including visual, textual, and mathematical formats. This allows for a thorough examination of how different input modalities affect model performance. Their findings reveal that across various foundation models, there's a consistent preference for textual representations over visual ones. For instance, when evaluated on IsoBench problems, models like Claude-3 Opus, GPT-4 Turbo, and Gemini Pro perform significantly worse when provided with images instead of text. Claude-3 Opus shows a decrease of 28.66 points, GPT-4 Turbo 18.71 points, and Gemini Pro 14.87 points. To address these performance gaps, the authors propose two prompting techniques: IsoCombination and IsoScratchPad. These techniques aim to improve model performance by considering combinations of different input representations and facilitating translations between them. Overall, this work sheds light on the importance of input modality in influencing the capabilities of foundation models and offers strategies to enhance their performance in diverse problem-solving scenarios.

**Questions To Authors:**

How many question do you have in total for the benchmark? Currently it is not clear from the paper.

**Reasons To Accept:**

The study is thorough, the conlcusion obtained is solid to me, which includes: Visual representations may be insufficient, Models cannot utilize low-level visual features for generation, Representation matters even for text. Both IsoCombination and IsoScratchPad are considered.

**Reasons To Reject:**

I think the overall quality is good, I didn't find a concrete reason to reject. I will listen to other reviewers opinion.

---

> ### Author Rebuttal · Authors · 2024-05-31
>
> Thank you for your positive feedback and comments. We are glad that you  find that IsoBench sheds light on the importance of input modalities and offers simple strategies to enhance their performance on image modalities. We are also grateful you find our study thorough, and the conclusions solid.
>
> ### **Number of questions in IsoBench**
> For the validation split of IsoBench, we have 1,887 samples in total. We will continue to enrich the diversity and scale of our datasets, and will open-source our dataset and code for data synthesis. We will further collect a test split and make it public soon.

---

> > ### Comment · Reviewer_R3GK · 2024-06-03
> > **New response**
> >
> > I don't have additional questions at the moment and will adjust my score according to discussion with other reviewers.

---

### Official Review · Reviewer_aqMK · 2024-05-04

**Rating:** 4
**Confidence:** 5
**Ethics Flag:** 1

**Summary:**

The paper proposes a benchmark, IsoBench, to assess the performance of the large multimodal models on the isomorphic representations of the input. For example, difference in the performance of the models on a task with image input versus the image caption (text) as an input. The paper finds that the LMMs perform way better for text representations than image representations. In addition, the paper proposes two methods IsoCombination and IsoScratchPad to improve the performance on their dataset.

Overall, I believe that the paper is not well-motivated and the main contribution is not interesting or surprising for the practitioners.

1. Primarily, I feel that the authors fail to understand that you make the task easier for a model when you represent an input as text instead of the image. Specifically, when you want to solve a task over an image input, the LMM has to correctly “perceive” objects, “read” text, “understand” layout and then perform relevant reasoning. For example, if an image has an embedded text “1+2=?” then the model needs to read the embedded text perfectly and then solve the problem of summation. Whereas, if you provide the text directly “1+2=?” as a prompt to the model, it just has to perform the reasoning part. Hence, it is expected that the model will be better on the text representation of the input instead of the visual representation of the input.
2. The introduction never justifies why their assessment is useful for the practitioners or for any relevant downstream task. “Testing whether the multimodal fusion components of these models truly empower the model with the same capabilities when reasoning about images and text.” is the effect of the dataset, but the motivation to study this remains unclear.
3. I do not like the connection with picture superiority effect. The way LMMs are designed (internet-scale data) is not the same as how humans function (limited knowledge but a lot of visual feedback in our daily lives).
4. I like the fact that the authors collected the dataset for a large range of domains (game playing and science) and did a good job at identifying various kinds of tasks within each domain.
5. IsoCombination is a contrived method which would not work in practice. Since their dataset already has isomorphic representations (either hand-written for science domain or using existing tools), it was easy for them to prompt GPT-4V with them. However, a user would not create isomorphic representations for the model.
6. Image captioning is a very popular VL task. I do not like the fact that the authors repackage it as something novel and call it IsoScratchPad. This method is popularly known as augmented LLM setup (see MathVista) where image captions are provided to the LLM. It is well known that it does not work when the generated captions are not accurate (e.g., MathVista). As a result, this approach is quite worse than providing the model with the entire image itself on complex reasoning tasks.

**Questions To Authors:**

Mentioned in the summary

**Reasons To Accept:**

Mentioned in the summary

**Reasons To Reject:**

Mentioned in the summary

---

> ### Author Rebuttal · Authors · 2024-05-31
>
> We thank you for your comments and suggestions, and we are glad to see the you appreciate that IsoBench covers a large range of domains and exemplary tasks.
>
> ### **On implicit bias (and level of difficulty) of different modalities**
> The implicit bias is indeed what we aim to show. Even though different modalities are all processed from their tokenizations, in most multimodal LLMs, there is still a sensible performance gap between them—as IsoBench shows. We agree that the current pretraining paradigm may favor textual representations, and this is exactly the point we’re trying to make, and IsoBench is designed to showcase this phenomenon and serve as a benchmark for future improvements.
>
> We respectfully disagree that tasks are easier with text inputs than with image inputs. The LLM must understand token relationships in text, similar to layout understanding in images. For instance, a chess board image and its Algebraic Notation Language (ANL) text representation both require the LLM to comprehend piece positions and relationships, involving similar reasoning processes.
>
> ### **Why IsoBench is useful for practitioners**
> The performance gap of visual inputs from that of text may point to a direction of improvement. Matching capabilities from visual representation to those from textual representation is a direction that can be pursued in the future by practitioners.
>
> ### **Connection with Pictorial Superiority effect**
> To clarify, we do not claim that VLMs should exhibit the same preference as humans. Nevertheless, (dis)analogies between neural models and cognitive science is common in ML literature (e.g. [1]), and we hoped that this example could help provide intuition.
>
> ### **IsoCombination and IsoScratchPad are trivial and contrived**
> We agree that IsoSP and IsoCB are simple strategies. This was our intention, and these methods are purposefully designed to be minimal to demonstrate possibilities for closing the vision-text gap. We also hope that others could use these initial strategies to develop more-sophisticated methods in future work.
>
> As Reviewer CR6d pointed out, in many scientific fields, the images are complicated, researchers usually provide extra text inputs to help the model. This is also a common practice in scientific fields, where figures and tables in most papers are coupled with captions, providing descriptions and takeaways. We will make such motivation clearer in our next revision.
>
> [1] https://arxiv.org/abs/1710.09829

---

> > ### Comment · Reviewer_aqMK · 2024-06-02
> > **Reply to Rebuttal**
> >
> > I thank the authors for their rebuttal. I have increased the score to 4 since the authors have done a decent job in arguing the contribution of the work. Specifically, I understand that the paper identifies the gap in the model’s understanding of an identical input when grounded in the image representation or the text representation. However, I believe that the paper and the rebuttal have not been able to justify the usefulness of the knowledge about this gap. In addition, I am not against the simplicity of the method, but I feel that the paper sufficiently shows its applicability for practical tasks.
> >
> >
> > ### **On implicit bias (and level of difficulty) of different modalities**
> >
> > I agree that a chess board image or an ANL representation require LLM to comprehend piece positions and relationships, involving similar reasoning processes. However, I still feel that the task of comprehending the piece positions and relationships is easier in the text modality. Assume that the location of the rook in Figure 2 is d8 – this location is explicitly mentioned in the text representation by three dots before ‘r’. The text position encodings can easily tell that the location of the rook is 4th from the left. In addition, the text representation explicitly mentions the character ‘r’ that disambiguates the presence of the rook itself.
> >
> > ### **Why IsoBench is useful for practitioners**
> >
> > While the benchmark shows that there is a gap between image and text representations, we should also acknowledge that the tasks studied in the paper are more commonly grounded in the “image representation” instead of the “text representation”. For instance, I will not know the underlying mathematical function for the graphs shown in Figure 2, and will not be able to write the text representation (function form) to solve my queries.
> >
> > ### **IsoCombination and IsoScratchPad are trivial and contrived**
> > I am fine with the triviality of the approach – it is okay if simple things work. However, the paper does not provide any guidelines on the scalability of the approach. The paper or the rebuttal does not tell me how to solve questions in SciBench or MathVista using IsoCombination. In practice, I will not be able to find an expert to write accurate text representation of the images.
> > In addition, I think image captioning is repackaged as IsoScratchPad. Infact, Section 3 argues that the “models only offer a coarse-grained representation that only summarizes high-level features, and may be unsuitable for detailed analysis such as plot and chart understanding.” and then ends up using the same models for image captioning (IsoScratchPad).

---

> > > ### Author Response · Authors · 2024-06-04
> > >
> > > Thank you for engaging in an in-depth discussion with us during the rebuttal period!
> > >
> > > ## **On implicit bias (and level of difficulty) of different modalities**
> > >
> > > > I still feel that the task of comprehending the piece positions and relationships is easier in the text modality. Assume that the location of the rook in Figure 2 is d8 – this location is explicitly mentioned in the text representation by three dots before ‘r’. The text position encodings can easily tell that the location of the rook is 4th from the left. In addition, the text representation explicitly mentions the character ‘r’ that disambiguates the presence of the rook itself.
> > >
> > > We just want to point out an interesting observation that we think gives some evidence against this point. In our experiments, the best-performing text representation was, in fact, PGN, which does not describe the board state nor locations of pieces (as in the other representations), but instead records a trajectory of moves from the start of the game. As an example, here’s the **PGN for the board in Fig. 2**: 1. e4 e5 2. Nf3 Nc6 $\cdots \cdots \cdots$ 27. Ke3 Qf3#. This representation is **harder than just getting the current board state** (as in ANL and image representations) as it must do state-tracking to infer the current position. This gives an example of how a more-explicit representation of the current board state, which seems intuitively easy to us, is not always the easiest for these models.
> > >
> > > ## **re: usefulness for practitioners**
> > >
> > > Thank you for this suggestion! We will add to our paper an in-depth discussion on the usefulness of IsoBench for practitioners. We share a curtailed version below.
> > >
> > > Regarding the graphs, there are many natural cases where a practitioner (e.g., a data scientist) starts with a symbolic (tabular) representation, and then may find it natural to convert it to a visual plot. In these cases, they have a choice of which representation to feed to the model, and our results tell them which is likely to be superior.
> > >
> > > Similarly, for finance users who need to work with Markov Chains – even though visualization as chains/graphs are more interpretable to humans – our results tell them it’s better to provide LLMs with transition matrices of the Markov Chains instead of their graphical visualizations.
> > >
> > > ## **re: no experts can write accurate text representations**
> > >
> > > We agree that there are tasks that are more naturally grounded in image than text, e.g. the case of analyzing scientific plots presented in papers. Our math tasks are deliberately designed to assess necessary properties for analysis automation, such as counting, curvature, and symmetry. The models’ inability to reliably perform on these tasks serves as a cautionary tale to warn practitioners from overtrusting VLMs to carry out these analyses without expert insight.
> > >
> > > ## **re: same model for image captioning**
> > >
> > > If an expert themselves is using the LLM to help automate tasks, they could add text descriptions as needed. It's valuable for them to know that this is likely to be helpful.
> > >
> > > On the curious case of the effectiveness of IsoSP, our hypothesis is that VLMs are better at *perceiving* than *reasoning* with visual modalities. High-quality, textual reasoning dataset are abundant (e.g. [1]), but their VL counterparts (e.g. [2]) are scarce. Thus, reasoning with imperfect captions may outperform reasoning with complete visual input, due to the lack of quality training for VL reasoning. We aim to investigate this hypothesis in future works.
> > >
> > > [1] Platypus: Quick, Cheap, and Powerful Refinement of LLMs, https://platypus-llm.github.io/
> > >
> > > [2] LLaVA Visual Instruct 150K, https://huggingface.co/datasets/liuhaotian/LLaVA-Instruct-150K

---

### Official Review · Reviewer_yceX · 2024-05-13

**Rating:** 6
**Confidence:** 5
**Ethics Flag:** 1

**Summary:**

This paper proposes an interesting benchmark called IsoBench, designed to evaluate multimodal foundation models on tasks with isomorphic representation. It provides key observations on existing models and introduces prompting techniques (IsoCombination and IsoScratchPad) to integrate input modalities effectively.

The experiments offer empirical findings of the capabilities of existing models when processing different modal inputs.

**Questions To Authors:**

How do we pinpoint the root cause of the observed model biases, considering the complex system design behind these API?

**Reasons To Accept:**

- The paper thoroughly discusses and examines existing multimodal foundation models, enhancing our understanding of their capabilities in various modality input scenarios. The experiments are solid. The findings in the paper are valuable for future research.

- The proposed benchmark could serve as a great resource for performance evaluation.

- It introduces two prompting methods: IsoCB, which assesses the combined effect of using multiple representations, and IsoSP, which translates visuals to text to solve tasks with text-only inputs. Both methods have shown improved performance.

- The paper is well-written and easy to understand.

**Reasons To Reject:**

- The paper assumes that multimodal foundation models should possess strong visual capabilities, akin to human cognition. However, the path toward Artificial General Intelligence (AGI) remains undefined regarding this aspect. That is, emulating human visual cognition could be advantageous but may also be limited by human capabilities. These models might surpass human cognition in certain areas that are beyond human perception. Thus, a preference for text representation is not inherently negative or positive. The focus should be on whether the models function effectively, irrespective of the representation type.

- While evaluating state-of-the-art commercial models like GPT-4V and Gemini is great, their performance may be influenced by several factors in the process chain, such as API calls, data transmission, prompt/image filtering for responsible AI, model inference, and output filtering. Each company's unique design behind these API calls makes it challenging to pinpoint the root cause of observed biases (i.e., preference for text inputs). The paper struggles to provide conclusive answers when considering the entire system design, which could limit its findings.

---

> ### Author Rebuttal · Authors · 2024-05-31
>
> Thank you for your positive feedback and suggestions. We are grateful that you recognize our efforts in curating the dataset, and find our analyses comprehensive and presentation clear. We are also glad that you find IsoBench useful to the broader multimodal community.
>
> ### **Preference for text representation is not inherently negative or positive**
> This is a great question! To clarify, we are not claiming that preferences toward a visual or textual representation are good or bad. The main observation is that multimodal foundation models exhibit preferences that are different from humans.
>
> Nevertheless, we believe multimodal models should possess strong visual reasoning knowledge capabilities, even if they cannot match pure textual inputs on certain tasks. Multimodal models can unlock a broad spectrum of applications in cases where visual inputs are critical and/or natural. Some examples include agents that navigate computer environments [1] and multimodal planning [2].
>
> ### **Root cause of the observed model biases**
>
> To our knowledge, this remains an open question, but one popular hypothesis is that multimodal models are endowed with a “language prior” [3] bias as a consequence of their training strategies.
>
> We would also like to use our observations from IsoBench to improve open-source multimodal models. One possible approach is to use mechanistic interpretability (e.g. [4]) to pinpoint the root cause (e.g. poor modal fusion, insufficient training FLOPs, etc) of the performance discrepancies. This would hopefully motivate better architecture design and/or training strategies.
>
> [1] OSWorld: Benchmarking Multimodal Agents for Open-Ended Tasks in Real Computer Environments, https://arxiv.org/abs/2404.07972
>
> [2] Fine-Tuning Large Vision-Language Models as Decision-Making Agents via Reinforcement Learning, https://arxiv.org/abs/2405.10292
>
> [3] Revisiting the Role of Language Priors in Vision-Language Models https://arxiv.org/abs/2306.01879
>
> [4] Interpreting CLIP's Image Representation via Text-Based Decomposition, https://arxiv.org/abs/2310.05916

---

> > ### Comment · Reviewer_yceX · 2024-06-03
> >
> > I have no further questions now, and I am still positive about this paper. I would like to keep my initial rating.

---

### Official Review · Reviewer_CR6d · 2024-05-24

**Rating:** 6
**Confidence:** 5
**Ethics Flag:** 1

**Summary:**

The paper introduces a new benchmark dataset, IsoBench, which aims to evaluate multimodal foundation models on problems with isomorphic representations. The authors have put effort into constructing the dataset and conducting evaluations on various models. The paper is generally well-structured and easy to follow, with the use of tables, figures, and examples to illustrate the findings.

While the work addresses an interesting research problem and provides some insights into the performance of multimodal foundation models, there are several concerns that limit its overall significance and originality. The task setting may not fully reflect real-world applications, and there are issues with data annotation and missing details that could affect the reliability of the benchmark. The proposed techniques, IsoCombination and IsoScratchPad, are relatively simple and do not provide substantial novel insights. Addressing these concerns and expanding the benchmark to include more challenging and diverse figures could strengthen the contribution of this work to the field of multimodal understanding and reasoning.

**Questions To Authors:**

The paper introduces IsoBench, an innovative benchmark dataset that sheds light on the performance of multimodal foundation models across various isomorphic representations. The authors' efforts in creating this dataset and conducting comprehensive evaluations are commendable and provide valuable insights for the research community.

However, to further strengthen the paper, I would encourage the authors to consider expanding the benchmark to include more challenging and diverse figures that better reflect real-world applications. Providing additional details on the data annotation process, such as the background of the annotators and quality control measures, would also help bolster confidence in the dataset's reliability.

Moreover, while the proposed IsoCombination and IsoScratchPad techniques offer interesting avenues for exploration, developing more advanced models that build upon the findings of this study could yield even more impactful insights.

**Reasons To Accept:**

The paper addresses an important research question in evaluating multimodal foundation models and provides valuable insights for future work.

The comprehensive evaluations and analysis reveal consistent preferences for textual representations and identify specific issues, such as visual recognition errors and insufficient fusion schemes.

The paper is well-organized, clear, and easy to follow, making it accessible to readers.

**Reasons To Reject:**

### (1) Task settings may not reflect real applications

For example, the science questions featured in IsoBench are relatively easy to describe in natural language. However, in real-world applications, many scientific figures are very complex and difficult to capture fully in natural language descriptions. These figures often serve as complementary information alongside the text. The benchmark may not adequately represent the challenges faced in real-world multimodal understanding tasks.

### (2) Data annotation concerns and missing details

The authors mention that a single author manually wrote descriptions for each figure in the science questions, with instructions to avoid introducing extra reasoning or information. However, there are several issues with this approach:
- Relying on a single annotator may introduce bias and inconsistencies in the data.
- Annotating high-quality descriptions for scientific images can be challenging and requires domain expertise. The paper does not provide information about the annotator's qualifications or background.
- There is no mention of any quality control measures or validation processes to ensure the accuracy and consistency of the annotations.

### (3) Missing data construction details

The paper lacks important details regarding the construction of the benchmark dataset. For example, the resolution of the image representations is not specified. Low-resolution images may result in poor understanding by current VLMs. Without this information, it is difficult to assess the validity and reliability of the benchmark.

### (4) Naive baselines

The two baselines proposed in the paper, IsoCombination and IsoScratchPad, are relatively simple and do not provide significant insights. It would be more valuable to see novel and insightful models that are inspired by the findings from the evaluations and analysis. The authors could explore more sophisticated approaches to address the performance discrepancies between different input modalities.

### (5) Related work

A superficial discussion on a very closely related work: MathVerse: Does Your Multi-modal LLM Truly See the Diagrams in Visual Math Problems?

---

> ### Author Rebuttal · Authors · 2024-05-31
>
> Thank you for the feedback. We are grateful that you appreciate our efforts in curating the dataset, and find our analyses comprehensive and presentation clear.
>
> ### **Task settings may not reflect real applications**
>
> We designed IsoBench for two main purposes:
>
> - **Our tasks are helpful predicates to real-world applications.**
> VLMs need to understand the node ordering (graphs) to analyze flowcharts; they need to capture curve patterns (math plots) to offer insights on data visualization. Our science tasks also serve as a first step for more complex real-world scenarios.
>
> - **A diagnostic tool for studying the difference across modalities is valuable to the community.**
> Most multimodal datasets require strong reasoning capabilities or knowledge to solve. It is challenging to understand why models perform poorly on these datasets. Simple tasks in IsoBench can help pinpoint failure modes on the visual component.
>
> ### **Data annotation concerns**
> We acknowledge the potential bias in using a single annotator. To address this, they were instructed to follow the guidelines emphasizing consistency and objectivity in description generation. Further, the authors conducted internal reviews of the annotations. We will include these detailed guidelines in the revision. Additionally, the annotator is a STEM PhD student with scientific domain expertise relevant to accurately annotating question.
>
> ### **Data construction details**
> We will add image resolution details to Appendix A, which contains details on dataset construction. We will open-source our datasets. Both graph and math problems are generated with 300 DPI in PNG. Chess and Science images are the same as Lichess and ScienceQA, respectively.
>
> ### **Naive baselines**
> IsoCB and IsoSP are designed to be simple to show existing VLMs’ strong bias towards “language priors” [2].
> Further, our main goal is to introduce IsoBench as a diagnostic tool for VLMs. We left developing sophisticated methods as future work.
>
> ### **Related work**
> We kindly want to note that we consider MathVerse [1] to be contemporaneous work, as it was released on arXiv on 03/21, and COLM submission deadline was 03/29.
>
> Nonetheless, we will include a more thorough discussion of MathVerse, which studies redundant, but not necessarily equivalent input representations on math problems. IsoBench studies the impact of isomorphic representations on a broader set of tasks.
>
> [1] https://arxiv.org/abs/2403.14624
>
> [2] https://arxiv.org/abs/2306.01879

---

> > ### Comment · Reviewer_CR6d · 2024-06-03
> > **Thank you for your comments**
> >
> > Thank you very much for your responses. You have addressed some of my concerns. I am pleased to increase the score.

---

### Decision · Program_Chairs · 2024-07-10

**Decision:**

Accept

**Comment:**

The paper introduces IsoBench to evaluate multimodal foundation models on problems with isomorphic representations.
Pro:
- The paper is well-written and clear
- The paper focuses on important research questions and provides comprehensive evaluation and benchmark

Cons:
- Most issues of missing details are relatively minor and can be handled in revision.
- There are some concerns about if the setting and assumptions are realistic and if there is new insight that can be learned from the findings. I would suggest the authors to address the comments in the revision.